# Association between Antibiotic Consumption and Resistance in Mink Production

**DOI:** 10.3390/antibiotics11070927

**Published:** 2022-07-09

**Authors:** Nanett Kvist Nikolaisen, Mette Fertner, Desiree Corvera Kløve Lassen, Chaza Nazih Chehabi, Amir Atabak Ronaghinia, Mariann Chriél, Vibeke Frøkjær Jensen, Lars Bogø Jensen, Karl Pedersen, Tina Struve

**Affiliations:** 1Research Group for Microbiology and Hygiene, National Food Institute, Technical University of Denmark, 2800 Kongens Lyngby, Denmark; chaza.chehabi@coopersurgical.com (C.N.C.); lboj@food.dtu.dk (L.B.J.); 2Department of Health and Diagnostics, Kopenhagen Fur a.m.b.a., 2600 Glostrup, Denmark; mefr@seges.dk (M.F.); amiratabak@sund.ku.dk (A.A.R.); tsvm@novozymes.com (T.S.); 3Center for Diagnostics, Technical University of Denmark, 2800 Kongens Lyngby, Denmark; dcokla@dtu.dk (D.C.K.L.); machr@mst.dk (M.C.); vifj@seges.dk (V.F.J.); 4Department of Veterinary and Animal Sciences, Faculty of Health and Medical Sciences, University of Copenhagen, 1870 Frederiksberg, Denmark; 5Department of Animal Health and Antimicrobial Strategies, National Veterinary Institute, 751 89 Uppsala, Sweden; karl.pedersen@sva.se

**Keywords:** antibiotic resistance, antibiotic consumption, *Staphylococcus delphini*, *Escherichia coli*, epidemiology

## Abstract

Antibiotic consumption is considered to be a main driver of antibiotic resistant bacteria. Mink breeding follows a distinctive seasonal reproduction cycle, and all of the mink produced in the northern hemisphere are bred, born, and pelted around the same time of year. Some of the diseases are age-related, which is reflected in the seasonal variation of antibiotic consumption. The seasonality makes mink a good model for the investigation of the association between antibiotic consumption and resistance. The objectives of this study were (1) to monitor the farm level of antibiotic resistance during one production cycle and (2) to assess the potential associations between antibiotic consumption and resistance. Twenty-four farms were included in this study (Denmark n = 20, Iceland n = 2, and The Netherlands n = 2), following a cohort of animals born in 2018. *Staphylococcus delphini* and *Escherichia coli* were isolated from samples of the carcasses and faeces and were collected randomly. The isolates were susceptibility tested and subsequently divided into the sensitive wildtype (WT) and the resistant non-wildtype (NWT) populations. The antibiotic consumption relative to the sampling periods was assessed as having a short-term or a long-term impact, i.e., in two explanatory factors. For both *S. delphini* and *E. coli*, a large between-farm variation of NWT profiles was detected. In the final multivariable, generalized linear mixed models, significant associations between NWT isolates and the consumption of specific antibiotics were found: the short-term use of tetracyclines in the growth period was associated with the occurrence of tetracycline NWT *E. coli* in the growth period (OR: 11.94 [1.78; 89.28]), and the long-term use of macrolide and tetracyclines was associated with the occurrence of erythromycin NWT *S. delphini* in the weaning period (OR: 18.2 [2.26; 321.36]) and tetracycline NWT *S. delphini* in the growth period (OR: 8.2 [1.27; 63.31]), respectively. Farms with zero consumption in the study years prior to sampling also had a substantial proportion of NWT isolates, indicating that NWT isolates are persistent and/or widely spread in the environment. Generally, a high occurrence of tetracycline NWTs was observed. NWT isolates with resistance against the most commonly used antibiotics were found on all the farms, stressing the need for routine surveillance and the prudent use of antibiotics. The results offer a preview of the complex relationship between consumption and resistance, demonstrating some significant associations between use and resistance. Moreover, antibiotic-resistant bacteria are present even on farms with no antibiotic consumption over extended periods, and theoretical explanations supported by the data are offered.

## 1. Introduction

The usage of antibiotics is considered to be the main driver of the development of antibiotic resistance. This applies to the antibiotic treatment of humans as well as animals and plants [1,2,3,4]. The antibiotic consumption and emergence of antibiotic resistance in livestock production has been given considerable attention in recent years due to the associated risk of treatment failure and the risk of transferring resistant bacteria between the animal and human reservoirs [5,6].

Global mink farming has been increasing for decades [7], and occasionally, the animals encounter bacterial diseases, calling for antibiotic treatment. Hence, the antibiotic consumption is linked to disease outbreaks caused by, e.g., *Arcanobacterium phocae* [8], *Clostridium septicum* [9], *Pseudomonas aeruginosa* [10], and infections with opportunistic pathogens (e.g., in relation to viral infections), i.e., *Escherichia coli* (may cause pneumonia, enteritis, and septicaemia, among others), *Streptococcus canis* (may cause pleuritis and wound infections, among others), or *Staphylococcus delphini* (may cause dermatitis, mastitis, and urinary tract infections, among others) [11,12,13].

Due to the mono-oestral reproduction biology of the species, mink production follows a distinctive production cycle [14]. In the northern hemisphere, the whelping of kits takes place from late April to early May, with pelting in November, and only the breeding animals are kept during winter for the forthcoming production year. About 10–20% of the farm capacity is kept as breeding stock. After mating in March, the males are pelted, and only female breeders are kept on the farm. Consequently, all the animals are exposed to the same time- and age-related risk factors throughout the production cycle. The management, feeding, and housing of the commercial mink are similar throughout Europe. The animals will only be exposed to latitude-dependent differences in climate by the semi-open or open housing. This provides a unique opportunity to measure seasonal variations of antibiotic consumption in one production cycle and assess the impact on antibiotic resistance, with the concurrent comparison of the antibiotic consumption in earlier production cycles.

The mink may encounter specific age-related problems, e.g., pre-weaning diarrhoea in the suckling kits [15], diarrhoea in the weaning period in May–June [16], or abortions due to *Salmonella* infection among the dams in the spring [17]. These disease outbreaks related to specific age groups are widespread and are reflected in the national antibiotic consumption patterns [12,16]. In the period after pelting, the health problems are rare and the antibiotic consumption during this period is therefore either very low or absent.

*Escherichia coli* and *S. delphini* are both commensals and some of the most important pathogens in farmed mink. *Escherichia coli* is considered a Gram-negative indicator species in many surveillance programs and is thoroughly investigated in many mammals and environments. Furthermore, the species has a zoonotic potential and a notable ability to transmit virulence and resistance genes [18]. The Gram-positive coccus *Staphylococcus delphini* is found as a commensal on the skin in both marine mammals (e.g., dolphins) and terrestrial mammals, including mink [19,20]. It has also been described in humans and, hence, has a zoonotic potential [21].

Associations between antibiotic consumption and the emergence of resistant pathogens have been described in other livestock, although in general the association is neither direct nor proportionate [22]. This association has never been assessed in the mink production. However, one study has indicated variations in resistance patterns amongst farms [23] and other described indications of the correlations between consumption and resistance patterns in clinical bacterial isolates from mink in Denmark [12]. The most commonly used antibiotics in Danish mink farms were aminopenicillins, tetracyclines, and macrolides [12,24]. The risk factors influencing antibiotic consumption (e.g., farm size, feed producer, veterinarian, and laboratory diagnostics) have been described in other studies [16,25].

Antibiotic resistance against tetracyclines can be mediated by several mechanisms [26]. Resistance genes have been identified in clinical *E. coli* isolates from mink: *tet*(A) and *tet*(B), these genes mediate an inducible efflux mechanism, and *tet*(M) in *S. delphini* isolates, which all mediate ribosomal protection [27,28]. Antibiotic resistance mechanisms against aminopenicillins are commonly the enzyme induction and secretion of beta-lactamases [29]. Beta-lactam resistance genes have been identified and are common in clinical isolates (*E. coli*, *S. delphini*) from mink [27]. One of three different mechanisms, rRNA methylation, efflux pumps, or enzymatic inactivation, may induce antibiotic resistance against macrolides [30]. Resistance genes belonging to the *erm* gene family (rRNA methylation) have been identified in clinical *S. delphini* isolates from mink [27].

Only one antibiotic drug (oxytetracycline) is registered for use in mink in Denmark [31]. So, in accordance with EU regulations [32,33], veterinarians are left to prescribe drugs that are registered for use in other livestock, such as pigs and cattle. The choices of antibiotic agents and dosages are mostly empirical and not, as they should be, based on treatment guidelines and a summary of product characteristics (SPCs) [33]. Inaccurate treatment of the animals may have consequences for animal welfare (a risk of, e.g., insufficient treatment and intoxication) and may lead to the emergence of antibiotic resistant pathogens.

We hypothesize that antibiotic consumption and the presence of resistant bacteria in mink farms are associated. The objectives of this study were (1) to monitor the farm-level antibiotic resistance during one production cycle and (2) to assess the associations between antibiotic consumption and resistance at the farm level.

## 2. Results

### 2.1. Antibiotic Consumption 2018

The majority of the antibiotics consumed in 2018 on the included Danish farms (n = 20) belonged to one of three antibiotic groups: penicillins, macrolides, and tetracyclines (Figure 1). The short-term antibiotic use (ABU) was assessed relative to the sampling time: (1) *Weaning ABU*: short-term antibiotic use from 16 April–30 June (bacterial sampling in June 2018). *Growth ABU*: short-term antibiotic use from 1 July–26 October (bacterial sampling in October 2018). *Breeding ABU*: short-term antibiotic use from 27 October 2018–25 March 2019 (bacterial sampling in March 2019). Long-term antibiotic use was defined as the consumption from 2016 up until sampling. The *weaning ABU* was primarily aminopenicillins. Only one farm had antibiotics prescribed in *breeding ABU* (Figure 1).

On the included Danish farms, the overall consumption across the farms decreased noticeably, by 50%, from 2017 to 2018 (2018: 1859.8 DADD/1000 biomass days; 2017: 3655.2 DADD/1000 biomass days) (DADD: defined daily animal dose, the assumed average dose needed to treat a 1 kg animal; biomass days calculated as in Jensen et al. [16]) (Appendix A Appendix A, Figure 2). On only four farms (12, 14, 18, and 19) did the antibiotic consumption increase in 2018. Four farms (4, 5, 8, and 16) had no antibiotic consumption in 2017 or in 2018. The remaining 12 Danish farms had a decreased antibiotic consumption. On the Icelandic and Dutch farms, the consumption of antibiotics in 2018 was in the lower 50th percentile of all the included farms (Figure 2).

### 2.2. Staphylococcus delphini

Regarding *Staphylococcus delphini* isolates, the statistical analysis comprised 543 isolates from Danish mink born in 2018, sampled during three sampling rounds from 2018–2019 (*weaning* (n = 225), *growth* (n = 161), and *breeding* (n = 157)) (Table 1). *Staphylococcus delphini* isolates (n = 55) from Icelandic mink born in 2018 and *S. delphini* isolates (n = 60) from Dutch mink born in 2018 were only applied for the assessment of non-wildtype (NWT) patterns and profiles (Appendix A).

In the Danish isolates, the proportion of benzylpenicillin and erythromycin NWT were lower than the proportion of tetracycline NWT in all three sampling rounds (Table 1). A statistically significant association between erythromycin NWT and the use of macrolides was observed for both the *weaning* and the *growth* sampling rounds (Table 2). Furthermore, a statistically significant association between the short-term use of tetracyclines and tetracycline NWT isolates was recorded for the weaning and the growth periods (Table 2).

The initial screening of predictors by univariable generalized linear mixed models resulted in the following predictors (*p* < 0.20) being included in the full models:Benzylpenicillin NWTs and the consumption of penicillins○*Weaning*: Long-term antibiotic use (*p* = 0.105);○*Growth*: No predictors;○*Breeding*: Short-term antibiotic use (*breeding ABU*) (*p* = 0.128);Erythromycin NWTs and the consumption of macrolides;○*Weaning*: Short-term antibiotic use (*weaning ABU*) (0.088), long-term antibiotic use (*p* = 0.009);○*Growth*: Short-term antibiotic use (*growth ABU*) (*p* = 0.128);○*Breeding*: No predictors;Tetracycline NWTs and the consumption of tetracyclines;○*Weaning*: No predictors;○*Growth*: Short-term antibiotic use (*growth ABU*) (*p* = 0.085), long-term antibiotic use (*p* = 0.030);○*Breeding*: long-term antibiotic use (*p* = 0.070).

The final multivariable generalized linear mixed models (Table 3) showed that the farms with the long-term use of tetracyclines (from 2016 up until sampling) (*p* = 0.030) were 8.2 times more likely to have tetracycline NWT *S. delphini* present in the weaning period (Table 3). The farms with long-term use of macrolides (from 2016 up until sampling) (*p* = 0.009) were 18.2 times more likely to have erythromycin NWT *S. delphini* in the weaning period (Table 3).

### 2.3. Escherichia coli

*Escherichia coli* isolates (n = 374) originating from Danish mink during two sampling rounds in 2018 (*weaning* (n = 194) and *growth* (n = 180)) were included in the statistical analysis. *Escherichia coli* isolates (n = 38) from Icelandic mink born in 2018 and *E. coli* isolates (n = 36) from Dutch mink born in 2018 were applied only for the assessment of the NWT patterns (Appendix A).

A low proportion of amoxiclav NWT was generally found in the two samplings (Table 4). Nevertheless, a statistically significant association was found between the consumption of penicillins and amoxiclav NWT in the *growth* sampling (Table 5). For ampicillin and tetracycline, a higher proportion of NWTs was found (Table 4). The highest proportion of NWT was for ampicillin in isolates from the *growth* sampling (Table 4).

The initial screening of the predictors by univariable, generalized linear mixed models resulted in the following predictors (*p* < 0.20) being included in the full models:Amoxiclav NWTs and the consumption of penicillins○*Weaning*: No predictors○*Growth*: No predictorsAmpicillin NWTs and the consumption of penicillins;○*Weaning*: Farm size (*p* = 0.161)○*Growth*: No predictorsTetracycline NWTs and the consumption of tetracyclines;
○*Weaning*: No predictors○*Growth*: Short-term antibiotic use (*growth ABU*) (*p* = 0.013), long-term antibiotic use (*p* = 0.193).

Subsequent backward elimination of the full models resulted in the final models, indicating that the farms using tetracyclines in the growth period were 11.94 times more likely to have tetracycline NWT *E. coli* isolated from faecal samples (*p* = 0.013) (Table 6).

### 2.4. Feed Producers

There was a significant association between the proportion of NWT isolates and the affiliated feed producer for all three types of NWT *S. delphini* isolates (*p* < 0.001) (Table 7). For *E. coli*, there was no significant association between the feed producer and the proportion of NWT isolates (Table 7).

### 2.5. Non-Wildtype Profiles of S. delphini

The most frequently isolated phenotypic NWTs expressed resistance to the antibiotic groups most commonly used, i.e., penicillins, tetracyclines, and macrolides (Table 8), followed by some of the historically widely used antibiotic drugs [12]: aminoglycosides, sulfonamides, and trimethoprim. Four isolates were NWTs when tested against cefoxitin.

A large variation of NWT profiles was detected (Denmark: 63 profiles, Iceland: 5 profiles, and The Netherlands: 11 profiles) (Appendix A). The isolates that were NWTs for ≥5 antibiotics represented 3% of all the isolates. No predominant profiles could be identified among the Danish isolates (NWT for ≥5 antibiotics n = 8). These multiresistant isolates originated from different farms (farm no. 5, 6 *, 10, 13 **, 15, where * were represented twice and ** were represented thrice), and the majority were isolated from the *weaning* sampling; one isolate was from the *growth* sampling; and one isolate was from the *breeding* sampling. These farms represented all three feed producers and different levels of consumption in 2018 (from 0–>300 DADD/1000 biomass days, Figure 2). The two farms in The Netherlands also had isolates that were NWTs for ≥5 antibiotics (n = 10). However, one of the farms only contributed with one of the isolates. When assessing the Dutch isolates, all the isolates had identical NWT profiles (benzylpenicillin, trimethoprim, sulfamethoxazole, and sulfamethoxazole in combination with trimethoprim and tetracycline), except for two isolates that were NWTs for an additional three antibiotics (erythromycin, streptomycin, tiamulin) (Appendix A).

All the Dutch *S. delphini* isolates (n = 60) were resistant to at least one of the tested antibiotics, whereas 84% (n = 456) and 36% (n = 20) of the Danish (n = 543) and Icelandic (n = 55) isolates, respectively, were resistant to at least one of the tested antibiotics. The isolates from the Icelandic farms had a lower proportion of NWT *S. delphini* compared to the isolates from the Danish farms. This may be a farm effect, as one Danish farm had a similarly low occurrence of NWTs. A larger number of Icelandic farms need to be investigated to determine if it represents a difference between countries.

### 2.6. Non-Wildtype Profiles of E. coli

The most frequently isolated phenotypic NWTs among the Danish isolates were against two of the three most used antibiotic groups in 2018, penicillins and tetracyclines (Table 9), but also some of the historically widely used antibiotic drugs [12]: trimethoprim, aminoglycosides, and sulfonamides. Twenty-four isolates had ASSuT (ampicillin, streptomycin, sulfonamide, tetracycline) NWT profiles (DK n = 18, IS n = 1, NL n = 5) [24] (Appendix A). Twenty-five isolates were 3rd generation cephalosporin NWTs (DK n = 17, IS n = 2, NL n = 6). Seventeen were NWTs for ceftiofur (DK n = 15, IS n = 1, NL n = 1) and eight isolates were NWTs for ceftiofur and cefotaxime (DK n = 2, IS n = 5, NL n = 1). Eleven isolates were colistin NWTs (DK n = 5, IS n = 2, NL n = 4).

Generally, the NWT profiles of *E. coli* from this study were very diverse (DK n = 43, IS n = 7, NL n = 24) and many isolates were NWTs for several antibiotics (Appendix A). The isolates that were NWTs for ≥7 antibiotics represented 5% of all the isolates. No common patterns could be identified among the Danish isolates (n = 11). These isolates originated from different farms (5, 6, 8, 9, 12, 14, 16 *, 18 *, 19, * represented twice), with all three feed producers, both samplings (*weaning*, *growth*), and different levels of consumption in 2018 (from 0–>300 DADD/1000 biomass days, Figure 2) being represented. One Icelandic farm and both farms from The Netherlands also had isolates that were NWTs for more than six antibiotics (IS n = 2, NL n = 7).

Sixty-six percent (n = 245), twenty-nine percent (n = 12), and seventy-eight percent (n = 78) of the Danish (n = 37), Icelandic (n = 38), and Dutch (n = 36) isolates, respectively, were resistant to at least one of the tested antibiotics

## 3. Discussion

### 3.1. Antibiotic Consumption

At a national level, the antibiotic consumption pattern in the Danish mink production changed in 2018, with a remarkable 40% reduction compared to 2017 [24]. This change was also notable on the 20 farms included in this study (Figure 2, Appendix A). The reduction may be ascribed to political attention from 2017–2018, which also resulted in a national antibiotic reduction action plan, enforced in autumn 2018 [34,35]. The preparation of this action plan took place in 2017 and 2018 in close collaboration between the affiliated government agency, the stakeholders, the veterinarians, the veterinarian association, and the mink farmers. Hence, the whole sector was focusing on antibiotic consumption. Yet, another potentially important factor in 2018 was the extreme weather condition (i.e., drought) during the whelping and weaning season, with the extremely dry conditions causing a reduction in the pathogen load on the farms. Nonetheless, the seasonal patterns of the antibiotic consumption corresponded with previous reporting, primarily penicillins in the weaning period, mixed consumption of antibiotic agents in the growth period, and minimal use after pelting [12]. At the farm level, the yearly variation in consumption is affected by problems occurring on the farms, such as variations in feed quality and disease outbreak. Disease outbreaks can be related to well-described, age-related problems, e.g., pre-weaning diarrhoea [15] or by the introduction of pathogens, including bacterial, e.g., *Pseudomonas aeruginosa* [10] or viral agents (e.g., mink enteritis virus [36]). In this study, four of the farms had an increasing consumption in 2018, and all four farms reported high mortality in the kits or in the young animals post-weaning. The introduction of pathogens could be related to the feed [37], or they could potentially be introduced when buying new breeding animals, by people visiting the farm, or by wildlife. Sixteen of the twenty Danish farms introduced new animals to their farms for the 2018 season.

The Dutch farms all had a lower consumption of antibiotics than most of the Danish farms (Figure 2), but several NWT isolates were recovered from the Dutch farms. The relatively low amount of in-feed antibiotics used at the Dutch farms was distributed to the animals over an extended period [38], meaning that the daily doses received by the animals were generally much lower compared to the other countries. Low dosages over extended periods are more likely to cause development of antibiotic resistance, compared to high doses for a short period of time [39]. Nonetheless, with the low representation of Dutch farms, it is not possible to draw any firm conclusions on these findings.

The studies on mink and other farmed species have reported that increased farm size is a risk factor related to antibiotic consumption [16]. In this study, only the binominal outcomes of WT/NWT of *E. coli* isolates against ampicillin were associated with farm size.

The use of non-antibiotic agents may select for resistance, e.g., zinc [40]. Seven of the included farms reported the use of zinc oxide topically on the kits and in the litterboxes in their efforts to treat pre-weaning diarrhoea. Methicillin-resistant *Staphylococcus aureus* (MRSA) isolated from mink carrying a zinc-resistance gene has been described [41]. Furthermore, this zinc-resistance gene is, in some MRSA linages, often carried along with tetracycline resistance genes [41,42]. The impact of the use of zinc on *S. delphini* isolated from mink is presently unknown. However, it is worth noticing that tetracycline NWT was the most frequently found NWT in the *S. delphini* (Table 8). Furthermore, it has been reported that tetracycline-resistant *S. delphini* isolated from mink carries the *tet*(M) gene [27]. In one of the farms using zinc, MRSA and cefoxitin-resistant *S. delphini* were isolated. In the antibiotic action plan of 2018, it was pointed out that the usage of zinc was to be terminated [34].

Few of the included farms had zero antibiotic consumption in 2018 (Figure 2). Still, a wide range of NWT isolates were found on these farms, which probably reflects that the NWT isolates were widely dispersed in all the habitats and emphasises the need for surveillance and the future prudent use of antibiotics. One Danish farm has had no antibiotic consumption at all since 2010, but multiresistant isolates were still recovered. In fact, from this particular farm, some of the NWT patterns included colistin and ciprofloxacin resistance in *E. coli* isolates and cefoxitin in *S. delphini*. This farm had no record of buying animals from other farms since 2011 (data retrieved from Kopenhagen Fur, April 2022).

### 3.2. Feed Producers

The significant association between the feed producers and NWT *S. delphini* reflects that either the NWT *S. delphini* is directly associated with the feed production, e.g., as in-feed contamination, or it is indirectly associated with geography as the feed producers were at three different locations in Denmark, e.g., for the local trade of breeding animals. There was no association with NWT *E. coli* and the feed producers. As *Staphylococcus delphini* is a commensal in mink [20], it is generally related to the animals and to a lesser extent to the environment, compared to the big pool of *E. coli* with a much greater range of potential hosts and environmental habitats.

### 3.3. Staphylococcus delphini

#### 3.3.1. Consumption vs. Non-Wildtypes

The association between a long-term use of tetracyclines and the occurrence of tetracycline NWT isolates in the growth period has not previously been documented on mink farms (Table 3). The association of a long-term use of macrolides and erythromycin NWT isolates in the weaning period has not previously been documented on mink farms (Table 3).

On the Danish mink farms, the second most prescribed antibiotic group in 2018 was macrolides, most frequently tylosin. Macrolides are active against Gram-positive bacteria, e.g., *S. delphini.* However, a recent study found that tylosin is not an appropriate drug of choice for treating extraintestinal infections in mink caused by *S. delphini* because a therapeutic concentration of the active compound was not achievable in mink plasma [43]. Consequently, the currently used off-label dose in Danish mink might cause more harm than good, considering that these sub-bactericidal doses may still select for antibiotic resistance—for the pre-existing resistant strains, but it may also promote de novo resistance [39].

In all three samplings, tetracycline NWT *S. delphini* isolates were found most frequently (54% of the Danish isolates, 293/543) (Table 8), corresponding to previous reports on antibiotic resistance patterns in staphylococci isolates from mink [11,12,41]. The NWT patterns in the random samples from this study give an indication of the general antibiotic resistance patterns found on the mink farms and a preview of the persistence of antibiotic-resistant bacteria on farms with no antibiotic use [44]. Earlier reporting on resistance levels was based on clinical isolates from mink, and in these cases, the animals might already have been treated with antibiotics [11,12]. Therefore, the level of antibiotic resistance based on the data from the clinical isolates most likely overestimates the general level of resistance in the population. Compared to the earlier reported MIC results on clinical isolates [12], the levels of NWT *S. delphini* isolates in these random samples were lower for two of the three most commonly used antibiotic classes (penicillins and tetracyclines). Only macrolides had a lower level of NWT in the earlier reported clinical isolates (22% NWT, 12/55) compared to the random samples (32% NWT, 172/543) from this study.

#### 3.3.2. Non-Wildtype Patterns

Four *S. delphini* isolates were NWTs when tested against cefoxitin, suggesting that these isolates might be methicillin-resistant. However, further investigations on these isolates revealed that none of them carried a *mecA* or *mecC* gene (data not shown).

Compared to the Danish and Dutch isolates, the Icelandic isolates had much simpler NWT patterns, being NWTs to a maximum of two antibiotics (Appendix A). The reasons for the differences in NWT prevalence between the countries are not clear and may simply represent a farm effect and not a true difference between countries. However, hypothetically, it may be the historically low antibiotic consumption in mink that is reflected in these NWT patterns. In 2018, the antibiotic consumption in the included Dutch farms was at low levels and comparable to that of Iceland. However, Iceland has for a decade been using very little antibiotics in the veterinary sector [32], and mink in Iceland has rarely been treated with antibiotics [45]. In The Netherlands, a wider palette of antibiotics has been used in the veterinary sector, particularly in cattle [46,47]. Thus, the environmental load of antibiotic resistance is likely to be much higher in The Netherlands. However, another factor to consider is that feed production in Iceland is on a very low scale compared to that of Denmark and The Netherlands. One of the Icelandic farms even produced fresh food at the farm site. The composition of the feed is, however, compatible with that of the other countries. Nonetheless, with the low representation of Dutch and Icelandic farms, it is not possible to draw any firm conclusions on these findings.

### 3.4. Escherichia coli

#### 3.4.1. Consumption vs. Non-Wildtypes

On the Danish farms, an association was found between the short-term use of tetracyclines in the growth period and the presence of tetracycline NWT isolates in the *growth* sampling (Table 6). We isolated these NWT *E. coli* from fresh faecal samples collected from the manure system. The samples originated from randomly selected cages representing each farm. This indicates that these NWT isolates are potentially selected by the use of tetracycline. However, our sampling method does not indicate whether the NWT *E. coli* were selected in the feed, in vivo, or later in the manure system. The tetracycline NWT *E. coli* found among isolates from the Icelandic and the Dutch farms were isolated from the faecal samples collected from the guts of the animals. Furthermore, a study found several tetracycline NWT *E. coli* isolated from the mink feed [48], which indicates that feed could be a route of introduction.

There is a potential risk of a spill-over of tetracyclines from the animals’ guts to the manure system, in which case *E. coli* NWT may be selected in the manure system rather than in vivo. The spill-over of antibiotics and the accompanying risks in all environments have been described [44]. The risk of spill-over in the mink production is even greater as there are no evidence-based dosages for mink. Consequently, there is a potential risk of overdosing, which would increase the spill-over. Additionally, the intestine of the mink is short, and hence, it has a very short time of passage of only 2–3 h [49]—this might also increase the amount of antibiotic spill-over to the manure system.

As *E. coli* is used as an indicator of antibiotic resistance, this NWT pattern potentially mirrors other NWT patterns in other bacteria. In Denmark, the use of tetracyclines has been the aim of attention as there is an official goal to reduce the high consumption of tetracyclines in the Danish veterinary sector [34,50]. As tetracyclines may select for other resistance patterns due to co-resistance [51,52,53], concerns should be raised about the use of tetracyclines in mink. With regard to that matter, a voluntary restriction on the use of tetracyclines was included in the antibiotic action plan of 2018 [34].

Compared to the earlier reported MIC results of clinical isolates [12], the occurrence of NWT isolates in these random samples was lower for two of the most commonly used antibiotic classes (penicillins and tetracyclines).

#### 3.4.2. Non-Wildtype Patterns

Twenty-four isolates had ASSuT NWT profiles. The ASSuT pattern of resistance has been described as the root of increased multiresistance in *Salmonella* in Europe [24,54]. This resistance pattern has been identified and described and is widely occurring in *E. coli* in domestic animals and humans. Due to the risk of the co-selection of resistance, it might explain how the tetracycline resistance is found to be consistently high in *E. coli* isolated from Danish pigs, even though the consumption of tetracyclines in pigs has decreased significantly since 2016 [24].

Isolates with reduced susceptibility (NWT) for 3rd generation cephalosporins (ceftiofur and/or cefotaxime) were identified. However, there were no reports on the use of cephalosporins in any of the included farms (DK, NL, IS). Generally, cephalosporins have not been used in Danish mink production for several years [35]. However, until 2010 the 3rd generation cephalosporins were widely used in the Danish pig production [55]. The presence of cephalosporin NWT in the mink production may be a result of spill-over from the pigs, either through the environment or from the use of feeding with porcine slaughter waste products. Documentation of the total and specified antibiotic use in past years in the Dutch and the Icelandic mink is not available. However, in The Netherlands, the consumption of cephalosporins in food-producing animals has been at very low levels for several years, and since 2019, no 3rd generation cephalosporin products have been available for food-producing animals [56]. In Iceland, in 2018, only 1% of all antibiotics sold for veterinary use were 1st and 3rd generation cephalosporins [57]. This indicates that the cephalosporin resistance has either been co-selected and preserved in the strains or maybe the strains have been introduced, e.g., with the import of animals from abroad. The latest import of animals to Iceland was in June 2015 from Denmark [58].

Isolates with NWT for colistin were also identified in this study. Whether these isolates carried any *mcr* genes was not investigated further. In Denmark, since 2017, the use of colistin in livestock has been phased out [24]. Colistin was not used in the veterinary sector in Iceland during the time period of 2014–2018 [57]. In The Netherlands, the sales of colistin in meat-producing livestock increased by 30% in 2018 [56].

## 4. Materials and Methods

### 4.1. Selection of Farms, Samples, and Bacteria

#### 4.1.1. Study Design

Twenty-four farms from three countries were initially included, 20 farms from Denmark, 2 from Iceland, and 2 from The Netherlands. These farms were typical farms for mink production in the northern hemisphere. Denmark was for many years one of the main contributors to the global mink production; hence, it was suitable to focus on these farms. The farms from Iceland and The Netherlands were, due to the low number of participating farms, only included as reference farms and for qualitative assessments. We performed a cohort study on the farms, where mink born in 2018 were followed for one production cycle, from whelping in April 2018 until mating in March 2019.

#### 4.1.2. Study Farms

With information from all 1455 active Danish farms in 2017 [59], the 20 enrolled study farms in Denmark were selected based on their total antibiotic consumption in 2017 (Appendix A Appendix A): high or low/no consumption. They were further stratified as all of the included farms were served from one of three feed producers located in three separate geographic areas, and they were selected so that the farms had animals with comparable genetic backgrounds, i.e., they had a mainly brown coat colour and/or a white coat colour. Some of the Danish farms withdrew from the study, and consequently, 20 farms participated in the first sampling, 18 farms participated in the second sampling, and 16 farms participated in the third sampling.

The two farms from Iceland and the two farms from The Netherlands were included as these countries represent consumption patterns that historically differ from those of Denmark. Historically, The Netherlands has had a more liberal policy regarding antibiotic consumption in the veterinary sector, whereas Iceland has had a generally low antibiotic consumption [47]. The farms were selected as representative farms for these countries. The selection was conducted in collaboration with the participating veterinarians and researchers in the respective countries.

#### 4.1.3. Sampling

The sampling periods were based on reports on tendencies in the antibiotic consumption on mink farms in Denmark. The samplings were conducted in relation to these described periods of more intense antibiotic therapy [12,16] (Figure 3).

Faecal samples and dead animals were randomly collected on the farms during each of the sampling periods for isolation of *E. coli* and *S. delphini*, respectively. The faecal samples were transported under cold conditions and stored at −20 °C. The dead animals were stored at −20 °C at the farm.

The short-term ABU during the periods preceding the three samplings was assessed. The three samplings were carried out as follows:(1)Weaning period*Weaning*: Kits (*S. delphini*) and faecal samples (*E. coli*) were collected in June 2018.*Weaning ABU*: Antibiotic use in the period from when the kits were born until sampling was completed in the weaning period: 16 April–30 June.(2)Growth period*Growth*: Young animals (*S. delphini*) and faecal samples (*E. coli*) were collected in October 2018.*Growth ABU*: Antibiotic use in the period from the previous sampling until the next sampling of young animals was completed later in the production cycle: 1 July–26 October.(3)Breeding period*Breeding*: Breeding animals (*S. delphini*) were collected in March 2019.*Breeding ABU*: Antibiotic use in the period from the previous sampling until the next sampling of breeding animals was completed at the termination of one production cycle: 27 October 2018–25 March 2019.

The animals from Iceland and The Netherlands were collected in the same periods as in Denmark. The animals were frozen at the farm site, shipped under cool conditions, received frozen in Denmark, and stored at −20 °C. During the necropsy of the Dutch and Icelandic animals, the faecal samples were all recovered from the aboral part of the intestines of the collected animals.

### 4.2. Bacterial Isolation

#### 4.2.1. *Escherichia coli*

The faecal samples were thawed at 5 °C overnight. The 10–15 mL faecal samples were diluted in 5 mL phosphate-buffered saline (PBS) buffer and homogenized. Subsequently, 100 µL of the suspensions were spread onto Columbia agar with 5% calf blood (Oxoid, Basingstoke, UK) and 100 µL on MacConkey agar (Oxoid) and incubated at 37 °C. The *Escherichia coli* colonies were identified by MALDI-TOF, as described by Nonnemann et al. [60].

#### 4.2.2. *Staphylococcus delphini*

The frozen cadavers were thawed, and the weight, age, and colour were recorded. Putrid animals were excluded as the putrefactive process and the microorganisms would hamper the bacterial isolation. Three samples were taken from each animal using sterile cotton swabs: (1) from the external acoustic meatus, (2) from the hairless interdigital area cranial to the metacarpal pads, and (3) from the left nostril. The third collection of samples was obtained on the farms from recently euthanized breeding animals. The anatomical sites mentioned above were swabbed using commercial sterile swabs transported in Amie’s medium with charcoal (Thermo Fisher Scientific, Waltham, MA, USA).

The samples were cultured on Columbia agar with 5% calf blood (Oxoid), incubated at 37 °C. The *Staphylococcus delphini* colonies were identified by MALDI-TOF [61].

### 4.3. Antibiotic Susceptibility Testing

The isolates were susceptibility tested in a semiautomatic system by broth micro-dilution (SensiTitre, ThermoFisher Scientific, East Grindstead, UK) in accordance with the methods described by the Clinical and Laboratory Standard Institute [61,62].

*Staphylococcus delphini* was tested against the following 14 antibiotics and ranges (DKVP SensiTitre panel): cefoxitin (0.5–32 mg/L), chloramphenicol (2–64 mg/L), ciprofloxacin (0.12–8 mg/L), erythromycin (0.25–16 mg/L), florfenicol (1–64 mg/L), gentamicin (0.25–16 mg/L), benzylpenicillin (0.06–16 mg/L), spectinomycin (16–256 mg/L), streptomycin (4–64 mg/L), sulfamethoxazole (32–512 mg/L), tetracycline (0.5–32 mg/L), tiamulin (0.25–32 mg/L), trimethoprim (0.5–32 mg/L), and sulfamethoxazole in combination with trimethoprim (19:1) (0.25–16 mg/L).

*Escherichia coli* was tested against the following 17 antibiotics and ranges (DKMVN4 SensiTitre panel): amoxicillin in combination with clavulanic acid (2:1) (amoxiclav) (2/1–32/16 mg/L), ampicillin (1–32 mg/L), apramycin (4–32 mg/L), cefotaxime (0.125–4 mg/L), ceftiofur (0.5–8 mg/L), chloramphenicol (2–64 mg/L), ciprofloxacin (0.015–4 mg/L), colistin (1–16 mg/L), florfenicol (2–64 mg/L), gentamicin (0.5–16 mg/L), nalidixic acid (4–64 mg/L), neomycin (2–32 mg/L), spectinomycin (16–256 mg/L), streptomycin (8–128 mg/L), sulfamethoxazole (64–1024 mg/L), tetracycline (2–32 mg/L), and trimethoprim (1–32 mg/L).

The isolates were distinguished as WT or NWT according to the available epidemiological cut-off values (ECOFF) and tentative ECOFFs (Appendix A Appendix A) [27,63]. The WT population consists of isolates that do not have phenotypically detectable resistance mechanisms, and the ECOFF is defined as the highest MIC in the WT population [63]. Hence, the NWT population consists of isolates with an MIC above the ECOFF and with a phenotypically detectable resistance mechanism. The identification of the proportion of NWT isolates is crucial in the surveillance of antibiotic resistance when there are no clinical breakpoints available. In this study, the NWT isolates were interpreted as an indication of the level of antibiotic resistance.

*Staphylococcus aureus* ATCC 29213 and *Escherichia coli* ATCC 25922 were included as quality control strains. A second investigator re-evaluated every 10th SensiTitre panel.

### 4.4. Antibiotic Consumption

When measuring antibiotic use, the Defined Animal Daily Dose (DADD) was applied. The DADD is defined as the assumed mid-range maintenance dose needed to treat a 1 kg animal. When measuring the antibiotic consumption, the seasonal changes in biomass were taken into account by applying 1000 biomass × day as the denominator, as in Jensen et al. [16], for each sample period (Figure 3). The unit, DADD/1000 biomass × days, describes antibiotic use relative to the biomass on the farm.

The consumption data from the Danish farms were extracted from VetStat [64,65]. The consumption data from the farms in Iceland and The Netherlands were based on information from the farmers and veterinarians; only the consumption data from 2018 were available. The consumption data defined the amount of antibiotics prescribed to the farms.

The Danish antibiotic use was evaluated long-term, using consumption data from 2016, 2017, and 2018.

### 4.5. Statistical Analysis

Due to a relatively low number of participating Danish farms (n = 20) with consumption of the antibiotics of interest (tetracycline, macrolides, aminopenicillins) in each of the respective periods (Figure 1), we dichotomized these variables as “use” versus “no use”.

For the descriptive statistics, a chi-square or Fisher’s exact test was applied (*p* < 0.05) to test whether the proportion of resistant isolates was associated with the usage of the respective antibiotic drug in the short-term period. Furthermore, these tests were applied to investigate whether the proportion of resistant isolates varied significantly between farm size and affiliated feed producer.

In terms of modelling, the outcome was NWT versus WT. For *S. delphini,* nine models were developed, covering three NWT profiles (benzylpenicillin, erythromycin, and tetracycline) in each of three sampling periods. For *E. coli*, six models were developed, covering three NWT profiles (amoxiclav, ampicillin, and tetracycline) and two sampling periods.

We applied a multivariable mixed effects logistic regression model at the isolate level to test for significant associations between NWT occurrence (outcome) and the predictors of interest. The assumptions for logistic regression models were met as the observations were randomly collected at the farm level, while the predictors were all categorized. The predictors included farm size (categorized), the use of the specific antibiotic drug in the short-term period (dichotomized), and the long-term use of the specific antibiotic drug (from 2016 until sampling in 2018) (dichotomized).

The farm sizes were categorized into small (<2500 breeding females), medium (2500–4000 breeding females), and large (>4000 breeding females) farms.

Initially, univariable, generalized linear mixed models were used to screen each predictor of interest. Factors with *p*-values *p* < 0.20 were subsequently included in the full model for the respective NWT profile and sampling period. Based on the full models, the final models were identified by means of backward elimination. The final model comprised the significant factors from the full model.

Univariable, generalized linear mixed models with binomial outcome (WT/NWT) of the Danish *S. delphini* and *E. coli* isolates against one of three antibiotics (*S. delphini*: benzylpenicillins, erythromycin, tetracyclines and for *E. coli*: amoxiclav, ampicillin, tetracyclines) were performed. All models included feed producer (n = 3) and farm (*S. delphini*: n = 20, 18, 16 and for *E. coli*: n = 20, 18) as random effects to account for the clustering of data. Antibiotic use describes whether the farm had used one of the affiliated classes of antibiotic drugs (either penicillins, macrolides, or tetracyclines) in the period just prior to sampling (*weaning ABU*, *growth ABU*, *breeding ABU*) or whether the use was long-term (from 2016 until sampling).

The variables for antibiotic use describe whether the farm had used one or more of the antibiotic drugs (either penicillins, macrolides or tetracyclines) in the same production cycle prior to sampling (*weaning ABU*, *growth ABU*, *breeding ABU*) or whether the use was long-term (from 2016 until sampling).

Data management and statistical analyses were carried out in R version 4.0.2 (R Core Team, 2020) and modelling based on the lme4 package [66].

## 5. Conclusions

To document a correlation between antibiotic resistance and antibiotic consumption is challenging. However, this study demonstrates statistically significant associations between the most commonly used antibiotics and resistance in both *E. coli* and *S. delphini*. Additionally, the significant associations involved both short-term antibiotic consumption (*E. coli* NWT and tetracycline) and long-term antibiotic consumption (*S. delphini* NWT and macrolides and tetracycline).

Mink served well as a study animal as it has a distinct and repetitive yearly production cycle in all countries in the northern hemisphere. Additionally, the farms contain a large number of animals, which allows a sufficient sample size.

In 2018, a remarkable decrease in the national antibiotic consumption in Danish mink farms was recorded. This is likely due to the massive attention on the antibiotic consumption in the veterinary sector and the implementation of an antibiotic action plan in 2018 in the mink production. The most commonly used antibiotic groups in 2018 on the Danish farms were macrolides, tetracyclines, and penicillins, as in the preceding decade.

Even farms with no antibiotic use within a three-year period had a diverse range of NWT patterns and profiles in *E. coli* and *S. delphini* isolates, some being NWT against up to ten different antibiotics.

Generally, the finding in this study emphasizes the importance of using laboratory diagnostics (identification and antibiotic sensitivity testing) to optimize the chance of therapeutic success—as NWT isolates against the most commonly used antibiotics are to some extent found on all farms. Additionally, the findings demonstrate the need for routine surveillance (optimally on farm level, nationally and internationally) and the prudent use of antibiotics.

The results from this study are relevant for other countries and animal production types. Additionally, they offer a preview of the relationship, though complex, of consumption and resistance, which is relevant to other animal productions.

## Figures and Tables

**Figure 1 antibiotics-11-00927-f001:**
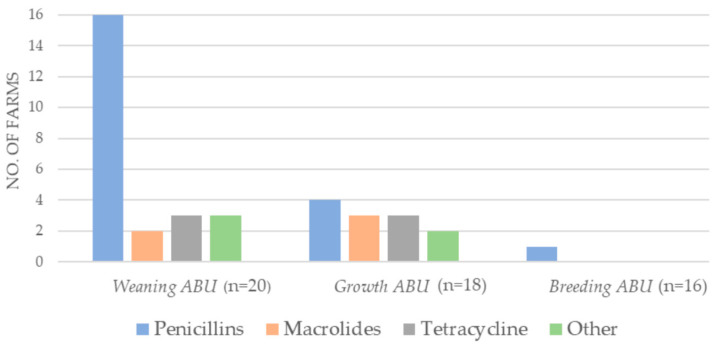
Antibiotic usage on mink farms (DK n = 20) during three short-term periods of the study period, 2018–2019. Number of mink farms with short-term consumption of the respective antibiotics (ABU), in the period just prior to the three sampling periods; *weaning ABU* (16 April–30 June), *growth ABU* (1 July–26 October), and *breeding ABU* (27 October 2018–25 March 2019). Other: Lincomycin in combination with spectinomycin; sulfamethoxazole in combination with trimethoprim, amphenicols, and aminoglycosides.

**Figure 2 antibiotics-11-00927-f002:**
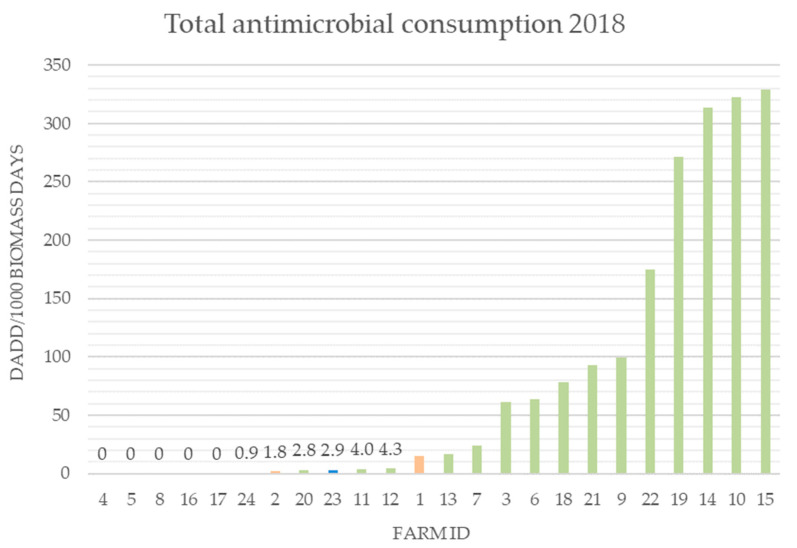
Total antibiotic consumption in 2018 on the included farms (DK n = 20, IS n = 2, NL n = 2). DADD: defined daily animal dose, the assumed average dose needed to treat a 1 kg animal; biomass days calculated as in Jensen et al. [16]. Farms 1 and 2 are Dutch (orange). Farms 23 and 24 are Icelandic (blue). Green farms are Danish.

**Figure 3 antibiotics-11-00927-f003:**
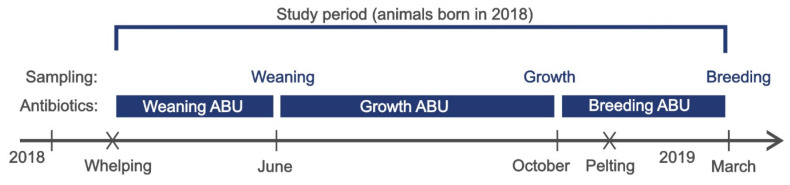
Timeline of the study and production cycle. Vertical lines indicate the sampling of animals and faeces. X indicates another important time point in the production periods: whelping in May and pelting in November. ABU: antibiotic use.

**Table 1 antibiotics-11-00927-t001:** *Staphylococcus delphini* isolates from Danish mink farms collected during three sampling rounds from 2018–2019.

Sampling Round	Number of Feed Producers	Number of Farms	Total Number of Isolates	Number of Isolates per Farm Mean (Range)	Proportion of Non-Wildtype Isolates at Farm Level Median [Q1; Q3]		
					Benzylpenicillin	Erythromycin	Tetracycline
*Weaning*	3	20	225	11 (10; 18)	0.20 [0.07; 0.40]	0.20 [0.00; 0.45]	0.35 [0.20; 0.85]
*Growth*	3	18	161	9 (4; 10)	0.20 [0.00; 0.30]	0.00 [0.00; 0.59]	0.65 [0.25; 0.90]
*Breeding*	3	16	157	10 (7; 10)	0.15 [0.00; 0.33]	0.20 [0.00; 0.63]	0.60 [0.28; 0.87]

*Weaning*: bacterial sampling in June 2018. *Growth*: bacterial sampling in October 2018. *Breeding*: bacterial sampling in March 2019.

**Table 2 antibiotics-11-00927-t002:** Antibiotic non-wildtype among *Staphylococcus delphini* isolates from Danish mink from 2018–2019. Association between antibiotic non-wildtype isolates and short-term use of the respective antibiotic drug on farm just prior to sampling (χ^2^-test results).

	Benzylpenicillin		Erythromycin		Tetracycline	
	Prevalence of Non-Wildtype	Association with PEN Use	Prevalence of Non-Wildtype	Association with MAK Use	Prevalence of Non-Wildtype	Association with TET Use
*Weaning*	52/225 (0.23)	*p* = 0.539	75/225 (0.33)	*p* < 0.001	108/225 (0.48)	*p* < 0.001
*Growth*	39/161 (0.24)	*p* = 0.105	41/161 (0.25)	*p* < 0.001	96/161 (0.60)	*p* = 0.006
*Breeding*	34/157 (0.22)	*p* = 0.340	56/157 (0.36)	No use	89/157 (0.57)	No use

*Weaning*: bacterial sampling in June 2018, antibiotic use 16 April–30 June. *Growth*: bacterial sampling in October 2018, antibiotic use 1 July–26 October. *Breeding*: bacterial sampling in March 2019, antibiotic use 27 October 2018–25 March 2019. PEN: penicillins, MAK: macrolides, TET: tetracyclines.

**Table 3 antibiotics-11-00927-t003:** Final multivariable, generalized linear mixed models with binomial outcomes of wildtype/non-wildtype of *Staphylococcus delphini* isolates, against one of three antibiotics (penicillins, macrolides, tetracyclines).

		Benzylpenicillin Non-Wildtype Isolates			Erythromycin Non-Wildtype Isolates			Tetracycline Non-Wildtype Isolates		
		*Weaning*	*Growth*	*Breeding*	*Weaning*	*Growth*	*Breeding*	*Weaning*	*Growth*	*Breeding*
Intercept Estimate (Std Error)		1.50 (0.40)	1.66 (0.46)	1.79 (0.49)	2.35 (1.31)	4.23 (3.27)	1.16 (1.28)	0.12 (1.17)	2.07 (0.00)	−0.18 (0.95)
Random effects (Std Dev)	Farm	1.25	1.39	1.41	1.65	3.14	2.54	1.47	1.35	0.59
	Feed producer	0.30	0.00	0.00	1.86	4.37	1.78	1.89	1.22	1.57
Short-term antibiotic use		-	-	-	-	-	-	-	-	-
Long-term antibiotic use	*p*-value	-	-	-	0.009	-	-	-	0.030	-
					0 (reference) ^1^				0 (reference) ^1^	
	Estimate (Std Error)				2.90 (1.12)				2.07 (0.00)	
	OR [CI95%]				18.2 [2.26;321.36]				8.2 [1.27;63.31]	

*Weaning*: bacterial sampling in June 2018, antibiotic use short-term, 16 April–30 June. *Growth*: bacterial sampling in October 2018, antibiotic use short-term, 1 July–26 October. *Breeding*: bacterial sampling in March 2019, antibiotic use short-term, 27 October 2018–25 March 2019. Long-term antibiotic use, from 2016 up until sampling. ^1^ No use of antibiotics has been used as reference: non-significant predictors not included in the final model.

**Table 4 antibiotics-11-00927-t004:** *Escherichia coli* isolates from Danish mink farms collected during two sampling rounds in 2018.

Sampling Round		Number of Farms	Total Number of Isolates	Number of Isolates per Farm Mean (Range)	Proportion of Non-Wildtype Isolates at Farm Level Mean (Range)		
					Amoxiclav	Ampicillin	Tetracycline
*Weaning*	3	20	194	9.7 (6; 10)	0.02 (0.0; 0.11)	0.25 (0.0; 0.70)	0.27 (0.10; 0.70)
*Growth*	3	18	180	9.5 (4; 11)	0.09 (0.0; 0.40)	0.47 (0.20; 0.90)	0.28 (0.00; 0.75)

*Weaning*: bacterial sampling in June 2018. *Growth*: bacterial sampling in October 2018. Amoxiclav: Amoxicillin in combination with clavulanic acid (1:2).

**Table 5 antibiotics-11-00927-t005:** Non-wildtype Escherichia coli isolates from Danish mink farms collected during two sampling rounds in 2018. Association between resistance and short-term use of the respective antibiotic drug on farm just prior to sampling (χ^2^/fisher-test results).

	Amoxiclav Non-Wildtype		Ampicillin Non-Wildtype		Tetracycline Non-Wildtype	
	Prevalence	Association with PEN Use	Prevalence	Association with PEN Use	Prevalence	Association with TET Use
*Weaning*	5/194 (0.03)	0.229	48/194 (0.25)	1	53/194 (0.27)	0.564
*Growth*	16/180 (0.09)	<0.001	85/180 (0.47)	0.433	47/180 (0.26)	0.244

*Weaning*: bacterial sampling in June 2018, antibiotic use 16 April–30 June. *Growth*: bacterial sampling in October 2018, antibiotic use 1 July–26 October. Amoxiclav: Amoxicillin in combination with clavulanic acid (1:2), PEN: penicillins, TET: tetracyclines.

**Table 6 antibiotics-11-00927-t006:** Final multivariable generalized linear mixed models with binomial outcomes of wildtype/non-wildtype of *Escherichia coli* isolates. Amoxiclav and ampicillin non-wildtype tested against the use of penicillins; tetracycline non-wildtype tested against the use of tetracyclines.

		Amoxiclav Non-Wildtype		Ampicillin Non-Wildtype		Tetracycline Non-Wildtype	
		*Weaning*	*Growth*	*Weaning*	*Growth*	*Weaning*	*Growth*
Intercept Estimate (Std Error)		3.63 (0.45)	2.66 (0.46)	1.20 (0.23)	0.18 (0.27)	0.99 (0.18)	−1.34 (0.39)
Random effects (Std Dev)	Farm	2 × 10^−7^	0.94	0.59	0.39	0.26	0.29
	Feed producer	0.00	0.00	0.00	0.32	0.00	0.25
Short-term antibiotic use	*p*-value	-	-	-	-	-	0.013
							0 (reference) ^1^
	Estimate (Std Error)						2.48 (0.92)
	OR [CI95%]						11.94 [1.78;89.28]
Long-term antibiotic use	*p*-value	-	-	-	-	-	-

*Weaning*: bacterial sampling in June 2018, antibiotic use short-term, 16 April–30 June. *Growth*: bacterial sampling in October 2018, antibiotic use short-term, 1 July–26 October. Long-term antibiotic use: from 2016 up until sampling. Amoxiclav: amoxicillin in combination with clavulanic acid (1:2). ^1^ No use of antibiotics has been used as reference: non-significant predictors not included in the final model.

**Table 7 antibiotics-11-00927-t007:** Antibiotic non-wildtype among *Staphylococcus delphini* and *Escherichia coli* isolated from Danish mink from 2018–2019. Association between the proportions of antibiotic non-wildtype isolates and affiliated feed producers (χ^2−^/Fisher’s exact-test results).

		FP A	FP B	FP C	*p*-Value
*S. delphini*	Tetracycline	0.13	0.85	0.36	<0.001
	Erythromycin	0.63	0.08	0.38	<0.001
	Benzylpenicillin	0.09	0.36	0.20	<0.001
*E. coli*	Amoxiclav	0.03	0.03	0.03	1
	Ampicillin	0.18	0.25	0.28	0.44
	Tetracycline	0.25	0.33	0.23	0.37

Amoxiclav: amoxicillin in combination with clavulanic acid (1:2), FP: feed producer.

**Table 8 antibiotics-11-00927-t008:** Non-wildtypes (NWTs) of *Staphylococcus delphini* isolates from the *weaning*, *growth*, and *breeding* samplings in Denmark (n = 543).

Antibiotic	(Antibiotic Group)	No. of NWT Isolates
Tetracycline	(tetracyclines)	293
Erythromycin	(macrolides)	172
Sulfamethoxazole	(sulfonamides)	135
Benzylpenicillin	(penicillins)	125
Streptomycin	(aminoglycosides)	108
Trimethoprim	(trimethoprim)	79
Spectinomycin	(aminocyclitol *)	25
Sulfa + TMP	(combination drug)	7
Tiamulin	(pleuromutilin)	6
Cefoxithin	(3rd gen. cephalosporin)	4
Florfenicol	(amphenicol)	3
Gentamicin	(aminoglycoside)	2
Chloramphenicol	(amphenicol)	2
Ciprofloxacin	(flouroquinolones)	1

*Weaning*: bacterial sampling in June 2018. *Growth*: bacterial sampling in October 2018. *Breeding*: bacterial sampling in March 2019. Sulfa + TMP: Sulfamethoxazole in combination with trimethroprim (19:1). * Spectinomycin is closely related to the aminoglycosides.

**Table 9 antibiotics-11-00927-t009:** Non-wildtypes (NWT) of *Escherichia coli* isolates from the *weaning* and *growth* samplings in Denmark (n = 374).

Antibiotic	(Antibiotic Group)	No. of NWT Isolates
Ampicillin	(penicillins)	133
Trimethoprim	(trimethoprim)	133
Tetracycline	(tetracyclines)	100
Streptomycin	(aminoglycosides)	100
Sulfamethoxazole	(sulfonamides)	95
Spectinomycin	(aminoglycosides)	47
Ciprofloxacin	((flouro-)quinolones)	47
Nalidixic acid	(quinolones)	26
Amoxiclav	(penicillins)	21
Chloramphenicol	(amphenicols)	20
Ceftiofur	(3rd gen. cephalosporins)	17
Florfenicol	(amphenicols)	5
Colistin	(polymyxins)	5
Neomycin	(aminoglycosides)	3
Gentamicin	(aminoglycosides)	2
Cefotaxime	(3rd gen. cephalosporins)	2
Apramycin	(aminoglycosides)	0

*Weaning*: bacterial sampling in June 2018. *Growth*: bacterial sampling in October 2018. Amoxicillin in combination with clavulanic acid (2:1).

## Data Availability

The data presented in this study are openly available at https://www.researchgate.net/publication/361861414_Cohort_study_Data_ABresistance_ABconsumption_2018-19 (accessed on 4 July 2022).

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
