# Peer review of "Association between Antibiotic Consumption and Resistance in Mink Production"

_antibiotics, 2022, doi:10.3390/antibiotics11070927_

Round 1

Reviewer 1 Report

This manuscript “Association between antibiotic consumption and resistance in mink production” prepared by Nikolaisen et al. presented the potential antibiotic treatment threats on mink farms. The topic of the manuscript is of great interest to the association between host antibiotic uptake and bacterial antibiotic resistance development. The mathematic models in analyzing antibiotic resistance are appreciated, and revisions clarifying the importance of the findings will be needed.

Major:

1.     In the introduction, I understand E. coli and S. delphini were used as model organisms to study host antibiotic input to bacterial antibiotic resistance development. However, the relevance of those two bacteria to mink disease is vague. Are they just taken as a representative for gram-negative/ gram-positive or they are related to stool pathogenic microbiome (E. coli) or skin pathogenic microbiome? How would it relate to diverse farms/minks in this study? 

2.     In the introduction, the mechanisms for different antibiotic resistance mechanisms need to be mentioned briefly, some of them can be acquired by outside environment treatment, and some of them are difficult.

3.     Among all farms included in the study, what are the differences among the mink species and the environment they farm the mink? Temperatures, other factors related to mink farming?

  1. Line 633, does the assumptions met for multivariable mixed effects logistic regression models, independence of errors, linearity in the logit for continuous variables, absence of multicollinearity, and lack of strongly influential outliers

  1. Line 635, does different mink types be considered random effects, does one farm only has one type of mink? 

Minor:

  1. The enhancing writing is in urgent need of better impact of this study. Some of the sentences are redundant and complicated, which leads to confusion.

  1. The bacteria species name should be italicized, please double-check through the manuscript.

  1. Significant association in several lines need to be indicated if it is statistically significant? If so what would be your cut off considering significance for those statistical tests.

  1. Line 131-133, Figure 2 figure illustration need to be adjusted to be together.

Reviewer 2 Report

Introduction

Please, summarise all the hypothesis in a paragraph at the end of the section, just before the objectives.

Results

2.1 the consumption of antibiotics across the farms: which farms exactly, those of the study or all farms of Denmark, please clarify.

Figure 2, please colorise.

All the tables are badly formatted.

I suggest to limit the presentation of results from Iceland and The Netherlands.

M & M

4.1 Please, how did you select the 20 farms in Denmark? Please list the criteria for inclusion.

4.3 Did you use the EUCAST criteria for resistance / susceptibility assessment?

4.5 please change risk factors to predictors, that is the correct word.

Discussion

The discussion includes findings, which is wrong, so please move these to results.

Overall. Useful manuscript that could potentially be accepted if the above changes are implemented correctly.

Round 2

Reviewer 1 Report

Thank you for addressing all the questions.

Author Response

Thank you for your constructive comments and critiques.

Reviewer 2 Report

1. Tables 2, 8, 9, 10 are still badly formatted (they must be aligned to the right part of the page).

2. I see the point regarding the international flare of the paper with Icelandic and Dutch farms, but really their number is so small, they are insignificant. The authors can leave them, but still this will be a mistake in the final manuscript. My strong suggestion is to exclude them, but the authors will take the responsibility for a future bad criticism.

Author Response

Thank you for your constructive comments and critiques.

The whole manuscript has now been revised with emphazis on the English language and style.

Point 1: Tables 2, 8, 9, 10 are still badly formatted (they must be aligned to the right part of the page)..

Response 1: All tables in the manuscripts has now been revised and modified.

Point 2: I see the point regarding the international flare of the paper with Icelandic and Dutch farms, but really their number is so small, they are insignificant. The authors can leave them, but still this will be a mistake in the final manuscript. My strong suggestion is to exclude them, but the authors will take the responsibility for a future bad criticism.

Response 2: We wish to keep the results from Iceland and the Netherlands, as the data indicate some interesting differences in NWT profiles. We have rewritten parts af the manuscript, to ensure that the Ducth and Icelandic data are interpreted according to the low number of enrolled farms. Additionally, we have emphazised the low number of farms. We have removed two tables with Icelandic and Dutch data, and mentioned the proportiens shortly in the text.